# Positive Correlation between Pesticide Consumption and Longevity in Solitary Bees: Are We Overlooking Fitness Trade-Offs?

**DOI:** 10.3390/insects11110819

**Published:** 2020-11-20

**Authors:** Verena Strobl, Domenic Camenzind, Angela Minnameyer, Stephanie Walker, Michael Eyer, Peter Neumann, Lars Straub

**Affiliations:** 1Institute of Bee Health, Vetsuisse Faculty, University of Bern, 3012 Bern, Switzerland; domenic.camenzind@vetsuisse.unibe.ch (D.C.); angela.minnameyer@vetsuisse.unibe.ch (A.M.); stephanie.walker13@gmail.com (S.W.); peter.neumann@vetsuisse.unibe.ch (P.N.); 2Laboratory of Soil Biodiversity, University of Neuchâtel, 2000 Neuchâtel, Switzerland; michael-eyer@bluewin.ch

**Keywords:** glyphosate-based herbicides, neonicotinoid, lethal and sublethal effects, combined exposure, *Osmia bicornis*

## Abstract

**Simple Summary:**

The possible impacts of neonicotinoids combined with glyphosate-based herbicides on bees are unknown. Here, we show no effects of chronic exposure to field-realistic dosages of Roundup^®^ and clothianidin alone or combined on food consumption and cumulative survival of adult female bees, *Osmia bicornis* in the laboratory. However, a positive correlation between exposure and longevity was revealed. Our data suggest a possibly neglected trade-off between survival and reproduction in insect toxicology.

**Abstract:**

The ubiquitous use of pesticides is one major driver for the current loss of biodiversity, and the common practice of simultaneously applying multiple agrochemicals may further contribute. Insect toxicology currently has a strong focus on survival to determine the potential hazards of a chemical routinely used in risk evaluations. However, studies revealing no effect on survival or even indicating enhanced survival are likely to be misleading, if potential trade-offs between survival and other physiological factors are overlooked. Here, we used standard laboratory experiments to investigate the sublethal (i.e., food consumption) and lethal (i.e., survival) effects of two common agricultural pesticides (Roundup^®^ and clothianidin) on adult female solitary bees, *Osmia bicornis*. The data showed no significant effect of the treatment on cumulative survival; however, a significant positive correlation between herbicide and insecticide exposure and age was revealed, i.e., bees exposed to higher dosages lived longer. As no significant differences in daily food consumption were observed across treatment groups, increased food intake can be excluded as a factor leading to the prolonged survival. While this study does not provide data on fitness effects, two previous studies using solitary bees observed significant negative effects of neonicotinoid insecticides on fitness, yet not on survival. Thus, we conjecture that the observed non-significant effects on longevity may result from a trade-off between survival and reproduction. The data suggest that a focus on survival can lead to false-negative results and it appears inevitable to include fitness or at least tokens of fitness at the earliest stage in future risk assessments.

## 1. Introduction

The ubiquitous application of agrochemicals is one major factor contributing to the ongoing decline in wild insect populations [1]. Indeed, recent pollinator declines, in particular numerous wild bee and butterfly species, have been closely associated with both single and combined exposure to pesticides [2,3]. Therefore, to determine the potential hazards of a xenobiotic, ecotoxicology aims to estimate the effects of chemicals on biological endpoints and represents an essential backbone for appropriate mitigation measures by stakeholders. However, current environmental risk assessment (ERA) appears outdated, flawed, and inadequate [4,5], with a strong emphasis on the measurement survival for non-target pollinator species, despite evidence of sublethal effects being more common and likely with equally severe consequences [6].

Sublethal effects induced by pesticide exposure can affect a wide spectrum of physiological as well as behavioral traits, including biochemical and enzymatic processes, neurophysiology (i.e., learning and orientation), ontogenesis, immune capacities, or reproductive traits [7]. In light of evolutionary biology, sublethal effects on fitness are likely to have detrimental consequences [8], and such effects are known to occur after neonicotinoid exposure [9,10,11]. Neonicotinoid insecticides and glyphosate-based herbicides are among the most common and widely applied pesticides in current agricultural practices [12], and are very likely to be found in combination [13].

Neonicotinoids act as an agonist of postsynaptic nicotinic acetylcholine receptors (causing a depolarizing blockade), which leads to paralysis and ultimately death of the affected organism [14]. Moreover, they are systemic and can thus be found in many aquatic- as well as terrestrial compartments, including agricultural soils (treated in preceding years), ecological flower strips, and organic agricultural soils [15,16,17]. Therefore, these broad-spectrum insecticides pose a high risk to non-target organisms [7], and unsurprisingly, numerous studies have demonstrated inadvertent negative sublethal and lethal effects for pollinators [18,19]. In contrast, while the mode of action is yet to be fully understood for insects, the herbicide glyphosate targets the shikimate pathway found in plants and microorganisms [20,21]. As animals lack the shikimate pathway, glyphosate-based herbicides are argued to represent one of the least toxic chemicals applied in agricultural practices [20,22]. However, increasing evidence reveals the contrary, as both lethal and sublethal effects have been revealed for invertebrates at field-realistic concentrations [23,24,25,26].

Despite the co-occurrence of neonicotinoids and glyphosate-based herbicides being virtually ubiquitous in agricultural landscapes, possible single and combined effects on behavior (e.g., food consumption) and survival are yet to be addressed. In this regard, food consumption rates are not only essential to calculate precise exposure scenarios, but may additionally provide insights on sublethal pesticide-induced behavioral changes [27,28]. For instance, the costs for detoxification processes can be compensated by increased food consumption [29], which may provoke potential trade-offs between detoxification and other biological functions [30,31]. Survival is a key endpoint measurement in test guidelines for non-target insect species at a Tier 1 level to assess an individual’s sensitivity toward a chemical. Furthermore, survival is used as a proxy to determine the lethality over time. Therefore, toxicological studies revealing no effect on survival or even indicating enhanced survival are likely to result in misleading implications [32], as sublethal effects can be overseen. Indeed, evidence revealing sublethal impacts yet no effects on survival exists [10,33]. Thus, studies on how combined pesticide exposure may affect lethal and sublethal endpoint measurements are required, to shed light on the likelihood of trade-off scenarios between detoxification and fitness.

Historically, honeybee (*Apis mellifera*) workers have served as a model insect to study the effects of pesticides [34], despite wild bee species representing the vast bulk of global bee biodiversity [35]. Moreover, eusociality plays a key role in the susceptibility of insects to environmental stressors due to superorganism resilience, which can be defined as the ability of colonies to tolerate the loss of somatic cells (=workers) as long as the germline (=reproduction) is maintained [36]. Hence, workers of any eusocial species will very likely differ in their responses compared to females of solitary species. Indeed, queens and workers of eusocial insects show differences in their ability to cope with neonicotinoids [37,38]. While ERA authorities acknowledged this critical point [39,40], the majority of published data still focus on Western honeybees, *A. mellifera,* and other bee species remain widely overlooked. This undoubtedly depicts a major knowledge gap, in particular, when considering that solitary bees of the genus *Osmia* are among the most pronounced generalist species [35], and are of high economic and commercial value for orchard pollination around the world [41]. While only a few studies have investigated the effects of combined pesticide exposure for solitary bee species (i.e., insecticide and fungicides), all indicating significant lethal or sublethal effects [28,33,42], similar effects of an insecticide in combination with a herbicide remain to be tested [43].

In a fully crossed laboratory experiment, we investigated the sublethal (i.e., food consumption) and lethal (i.e., survival) effects of two common agricultural pesticides in field-realistic concentrations on adult female solitary bees, *Osmia bicornis*. Using pre-established methods [28,44], we chronically exposed newly emerged individuals for 10 days to field-realistic, sublethal concentrations of insecticide (clothianidin) and glyphosate-based herbicide (Roundup^®^) singly or combined. By recording consumption we calculated corresponding exposure rates and assessed possible behavioral changes [27,28]. Furthermore, we assessed individual survival to determine the potentially lethal effects of the applied pesticides over time. Based on data from previous studies [28,33], we hypothesize that females exposed to single and combined pesticide treatments will experience significant lethal (i.e., decreased survival) and sublethal effects (i.e., suppressed food consumption) when compared with controls.

## 2. Materials and Methods

### 2.1. Experimental Set-Up

In June 2019, female *O. bicornis* (N = 300) from wild populations maintained in organic orchards (WAB Mauerbienenzucht, Konstanz, Germany) were placed in an insect rearing cage (BugDorm (47.5 cm^3^)) under complete darkness and room conditions (24 °C, 50% RH) until adult emergence. To establish known age-cohorts as well as to account for physiological and endogenous factors, emergence time and emergence mass (nearest 0.1 mg using an analytic scale (AT400, Mettler Toledo, Inc., Columbus, OH, USA)) were recorded daily over four days once the first cocoon eclosion was observed. Females that emerged after day four were excluded from the experiment (N = 82), thus resulting in a total of 180 emerged females. Cocoons that revealed no signs of eclosion after three weeks were cut open to determine the death of the bees (N = 38). This observed mortality rate of 12.7% lies within the range of previous reports for the genus *Osmia* [45,46]. Bees from each age-cohort (day 1–4) were then evenly distributed and randomly assigned to one of four treatment groups (N = 45 each): controls (pesticide-free), herbicide (Roundup profi^®^), insecticide (clothianidin), or combined (both Roundup profi^®^ and clothianidin). Bees were kept individually in separate cages (80 cm^3^) maintained at 24 °C, 50% RH, and indirect natural light with access to ad libitum sucrose-solution in 5 mL syringes [44]. Individuals were chronically exposed to their respective treatments for 10 consecutive days. Under the given conditions (i.e., indirect light exposure and constant room temperature) both chemicals are considered highly stable and reveal high persistence and degradation times [22,38,47]. Post-exposure the feeders were replaced with pesticide-free 50% sucrose-solutions (weight:weight (w:w)) and thereafter exchanged upon the sign of fungal contamination or at the latest every 10 days. The cage assay was terminated when the last individual had died.

### 2.2. Pesticide Solutions

Stock solutions of both clothianidin (99% purity, Sigma-Aldrich^®^, Buchs, Switzerland) and the commercial formulation of glyphosate (Roundup profi^®^, 480 g L^−1^ active substance, Landi AG^®^, Dotzigen, Switzerland) were prepared by dissolving the substances in distilled water (and acetone for clothianidin) at a nominal concentration of 1000 mg L^−1^, which was then diluted to 1 mg L^−1^. The stock solutions were then diluted added into a 50% (w:w) sucrose-solution to obtain field-realistic concentrations of 1.5 ng g^−1^ clothianidin (Insecticide) and 1.0 × 10^7^ ng g^−1^ glyphosate (Herbicide) [48,49]. The Combined treatment consisted of a mixture of the two pesticide solutions at the above-mentioned concentrations. Lastly, the final concentration of acetone (<1%) in the treatment solutions was also mixed in the Control sucrose-solution (50% (w:w)).

### 2.3. Endpoint Measurements

Mortality was recorded daily until all bees had died. To record sucrose-solution consumption and calculate individual exposure rates, feeders were weighed at day zero, day 10, or on the day of bee death during exposure. Using the mortality and consumption data obtained within the exposure period, the relative daily consumption (i.e., the weight of consumed sucrose-solution per weight unit of the animal per day (g × g^−1^ × day^−1^) of each bee was determined. Due to the known difficulties in feeding individual solitary bees under laboratory cage conditions [33,50] we opted for a conservative approach of excluding bees that may have died due to starvation. Therefore, we calculated the average relative daily consumption of sucrose-solution for the control bees and set the minimum consumption threshold at their lower 5th percentile. Bees from controls and treatment groups that consumed less than this value throughout the exposure period were thereby considered ‘non-feeders’ and were excluded from further analyses. Individual exposure rates were calculated by multiplying the weight of consumed sucrose-solution (i.e., g) by the concentration of the respective pesticide (i.e., ng g^−1^).

### 2.4. Statistical Analysis

The Shapiro-Wilk test and Levene’s test were used to test data and model residues for normal distribution homogeneity of variances and choose statistical tests, accordingly. Multilevel generalized logistic or linear regression models (GLMMs) with random intercepts were fitted using STATA16 (Stata Statistical Software: Release 16 (2019), StataCorp. LP, College Station, TX, USA), wherein individual bees were considered independent units, and treatments were included as the fixed term. Emergence time and mass were included as random effects as both factors can determine physiological conditions and consumption of bees [28]. Whenever possible, every multi-level model was compared with its single-level model counterpart. Therefore, a stepwise backward elimination approach was applied to determine the model of best fit for each multiple regression model using either a likelihood ratio (LR) test or the Akaike information criterion (AIC) and the Bayesian information criterion (BIC), using the functions *lrtest* and *estat ic*, respectively.

Emergence mass (mg) and relative daily consumption (g × g^−1^ × day^−1^) were non-parametrically distributed (Shapiro-Wilk’s test *p* < 0.05), so a Generalized Linear Mixed-effects Model (GLMM) was fitted using the *meglm* function corrected for Gamma distribution, to test for differences among treatment groups. Survival data were analyzed using the *mestreg* function for multilevel survival models in STATA16 at day 10 (i.e., post-exposure) and at the end of the experiment (i.e., after all bees had died). Median longevity was calculated at the 50th percentile of the survival time. Post-hoc comparisons for all variables among treatment groups were conducted using a multiple pairwise comparisons test (Bonferroni Multiple Comparisons Test (*bmct*)), defined by the function *mcompare(bonferroni)*. A logistic GLMM was applied to test for treatment differences for the binary outcome variable non-feeders using the function *melogit*, whereby the conditional distribution of the regression given the random effects was considered to be Bernoulli. Lastly, possible relationships among measured variables (i.e., emergence mass, emergence time, relative daily consumption, and longevity) were assessed by applying linear regression models using the function *regress*, wherein Treatment was included as a fixed factor and emergence time and mass as random effects. Whenever appropriate, either the arithmetic means ± the standard error (SE) or medians ± 95% confidence intervals (CI) of non-transformed values are given in the text. All statistical figures were created using NCSS20 (NCSS, LLC. Kaysville, UT, USA, ncss.com/software/ncss).

## 3. Results

No significant correlation was observed between emergence mass and emergence time (*F*_1,178_ > 0.02, R^2^ < 0.001, *p* > 0.89). Emergence mass among treatment groups did not significantly differ before exposure initiation (*meglm*, χ^2^ = 4.58, *p* > 0.20). Relative daily consumption during the exposure period did not significantly differ among treatment groups (*meglm*, χ^2^ = 1.29, *p* > 0.73; Figure 1A). However, emergence mass had a significant positive effect on consumption with heavier bees consuming more sucrose-solution (*z* = −2.18, *p* < 0.03). Average relative daily consumption across all treatments was 0.73 ± 0.65 − 0.79 (g × g^−1^ × day^−1^) (median ± 95% CI of median), with the lower 5th percentile being 0.1376 (g × g^−1^ × day^−1^). Therefore, a total of 19 individuals were revealed to have consumed below the set minimum consumption threshold and were subsequently considered to be non-feeders (Appendix A). The data revealed no significant difference in the proportion of non-feeders among treatment groups (*melogit*, χ^2^ = 6.15, *p* > 0.10), revealing 4.5–17.5% of the bees being non-feeders (Appendix A). Subsequently, pesticide-fed bees were exposed to either 1.08 ± 0.92 − 1.23 (ng × g^−1^ × bee^−1^) clothianidin (Insecticide) or 7.7 × 10^6^ ± 0.58 − 0.92 × 10^6^ (ng × g^−1^ × bee^−1^) glyphosate (Herbicide) or to both concentrations (Combined) (median ± 95% CI).

The following statistical analyses were conducted using only individuals that were considered ‘feeders’, resulting in the following sample sizes: N_Control_ = 43; N_Herbicide_ = 39; N_Insecticide_ = 42; N_Combined_ = 37. No significant difference in survival was observed among treatments after the exposure period at day 10 (χ^2^ = 5.20, *p* > 0.39) with neither emergence time nor mass with a significant effect (*z* < 1.16, *p* > 0.24). Likewise, no significant differences were observed among treatments when analyzing survival at the end of the experiment after all individuals had died (χ^2^ = 0.79, *p* > 0.85; Figure 1B) where median survival across all treatments was 21 ± 18 − 24 (d) (median ±95% CI). Emergence time however did have a significant effect on survival (*z* = −2.56, *p* = 0.01), where bees with later emergence times displayed prolonged longevity. Lastly, for all treatment groups, a significant positive correlation was observed between relative daily consumption within the first 10 days and longevity (*regress*, R^2^ = 0.24, *p* < 0.003; Figure 2); wherein the correlation did not significantly differ among treatments (*bmct*, χ^2^ = 51.05, all *p*’s = 1.00). Ultimately, an increased relative daily consumption consequently corresponds to an increased pesticide exposure.

## 4. Discussion

The data clearly show a significant positive correlation between pesticide exposure and longevity of adult solitary bees, *Osmia bicornis*, i.e., bees exposed to higher dosages, were surprisingly more likely to live longer. However, this had no significant effect on the overall cumulative survival of the treatment groups when compared to controls. While no significant differences in feeding behavior and relative daily consumption were observed across treatments, a significant positive correlation between increased food intake during the first 10 days (i.e., exposure period) and longevity was observed for all treatments and controls. While we have no data to support that fitness may be affected, previous studies using comparable concentrations of neonicotinoid insecticides have revealed a reduced reproduction, but no effects on adult survival [10,11]. Therefore, we postulate that exposed bees may opt for improved longevity over reproductive capacities (i.e., fitness) [28,51]. This may render substantial negative consequences and may provide a mechanistic explanation for previously observed negative effects of neonicotinoids on a population level in field studies [52,53]. The data further highlights that false-negative results may arise if fundamental mechanisms, such as detoxification abilities and/or trade-off scenarios are not accounted for when assessing survival measurements in toxicology. While a shift in current environmental risk assessments indeed seems long overdue to enable adequate policymaking [4,5], we urge future toxicological studies to focus on measures of fitness if we aim to effectively mitigate the evident role of agrochemicals on the ongoing mass extinction of species [8].

The data revealed no significant differences in the proportions of non-feeding behavior among all treatment groups. Furthermore, the daily consumption rate among the treatment groups did not significantly differ. As the nutritional status of a bee can alter its detoxification abilities and xenobiotic tolerance [54,55], we can exclude the possibility of caloric restriction (i.e., malnutrition) and starvation [56], which may have led to an immune deficiency [57]. Sublethal effects of pesticide exposure on consumption are often inconsistent across as well as within studies [27,58]. This may be due to varying dosages and/or modes of actions of active ingredients, as well as interspecific and intraspecific species differences [59,60]. Nevertheless, estimates on food consumption are fundamental to enable the calculation of exact pesticide exposure rates during chronic feeding tests as well as for identifying possible sublethal effects on behavior or metabolization. Indeed, biosynthesis and detoxification require significant energetic costs that can be compensated by increasing consumption [29], thereby ultimately masking the potential negative effects of a stressor [61]. Thus, measuring consumption rates may help understand possible energetic trade-offs between detoxification capacity and survival or reproduction [30,31]. Yet, as the data revealed no significant difference in consumption and insect detoxification comes at an energetic cost [29,62], the allocation of resources to chemical degradation may come at the expense of other functions [51,61,63]. While the detoxification pathways in bees are well documented [64,65]; the associated direct and indirect costs remain to be addressed.

There was no significant difference in survival among pesticide treatments and controls. While previous studies have indeed revealed lethal effects of single and combined pesticide exposure, these differences may again be due to varying pesticides and their combinations (insecticide and herbicide vs. insecticide and fungicide [28]), different modes of action (inhibitor of the shikimate pathway vs. inhibitors of ergosterol-biosynthesis [42]), differing exposure routes and their concentrations (sucrose-solution vs. sucrose-solution and pollen; 1.5 ng g^−1^ vs 8.6 ng g^−1^ [28]), as well as the diverse levels of susceptibility across and within a species (e.g., solitary bees vs. eusocial stingless bees [24], diploid females vs haploid male drones [19], newly emerged bees vs. elder bees [66]). In line with our study, neonicotinoid exposure at similar concentrations in *O. bicornis* revealed no effects on survival, yet showed drastic sublethal effects (i.e., suppressed syrup consumption and decreased thoracic temperature after imidacloprid exposure [33]), as well as decreased reproduction and a male-biased offspring sex ratio after combined thiamethoxam and clothianidin exposure [10]. Comparative survival data for glyphosate-based herbicides are currently lacking, yet past studies suggest that the commercial formulations are evidently far more toxic due to the added adjuvants inert diluents (e.g., ethoxylated alkylamines) [67]. Nevertheless, studies have revealed sublethal effects of the active ingredient glyphosate on beneficial gut microbiota, behavior, and memory in honeybees [23,25,26]. Additional studies are required to determine field-realistic exposure scenarios of glyphosate as well as the adjuvants in the commercial product for insect species.

While increased consumption during the exposure period across all treatments had a positive effect on the outcome of an individual’s longevity, this result does come surprisingly for the pesticide-exposed treatments. Consequently, a significant positive correlation between pesticide exposure and longevity was revealed, suggesting that individuals exposed to increased doses of pesticides were likely to live longer. While we can only postulate, this counterintuitive result may be explained by the common phenomena of pesticide-induced hormesis [32]. Hormesis is a biphasic dose-response whereby exposure to stress (e.g., pesticides) at low doses can stimulate biological processes [68], resulting in apparent misleading beneficial effects on endpoint measurements (e.g., survival). Irrespective of the underlying mechanism, the consequence of increased survival is likely to trade-off for other physiological factors [69], possibly affecting fitness [30,63]. Indeed, ample evidence exists from the laboratory and the field indicating such trade-off scenarios, wherein pesticide exposure decreased fitness parameters while not necessarily affecting survival [10,11,70]. Undoubtedly, our experiment would have benefited from directly measuring detoxification capacities and fitness (i.e., production of offspring) or sublethal parameters that adequately reflect fitness (i.e., ovary development in females or sperm quality in males) to confirm a possible trade-off scenario in solitary bees. If such a trade-off scenario is indeed confirmed this would be a novel mechanistic explanation for recently observed wild insect population declines [71,72].

Lastly, irrespective of the treatment group, both emergence mass and emergence time significantly influenced endpoint measurements, whereby a positive correlation was observed between emergence mass and consumption, as well as between emergence time and survival. These results underline the importance of considering physiological and endogenous factors (e.g., emergence time, emergence mass, age, or genetics) for toxicological studies, as they may alter the response to pesticide exposure [10,28]. Thus, it is of high relevance to consider such factors when agreeing upon standardized test guidelines for solitary bees that are currently under consideration and development.

## 5. Conclusions

Ultimately, the data as well as previous findings suggest that measuring survival for determining the risk imposed by pesticides may lead to inadequate conclusions if sublethal parameters, as well as underlying biological mechanisms, are neglected. As fitness is ultimately the key factor governing all wild populations, it appears inevitable to include fitness or at least tokens of fitness at the earliest stage in risk assessments (i.e., Tier 1) to avoid false-negative results. Therefore, it is of urgent relevance that sufficient and appropriate test guidelines for solitary bees are agreed upon to enable collecting reliable and replicable data. In conclusion, we urge future ERA to shift focus to ecotoxicological studies measuring fitness [8], if we aim to mitigate the effects of agricultural chemicals and protect natural biodiversity.

## Figures and Tables

**Figure 1 insects-11-00819-f001:**
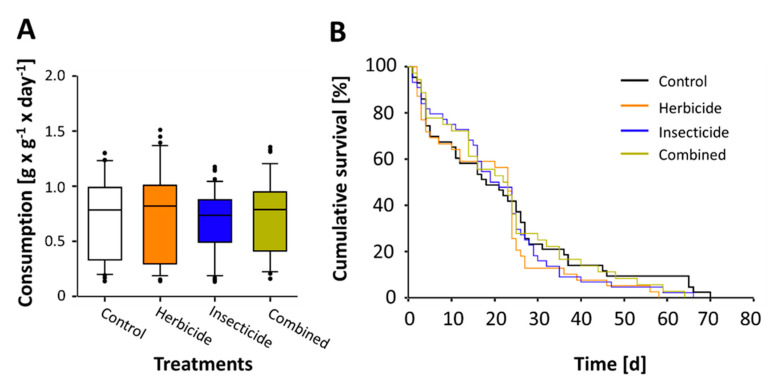
Consumption and cumulative survival of female adult solitary bees (*Osmia bicornis*). (**A**) Relative daily consumption measured during the 10-day exposure period revealed no significant differences among treatment groups (*meglm*, *p* > 0.73). The boxplots show the inter-quartile range (box), the median (black-line within the box), data range (horizontal black lines from the box), and outliers (black dots). (**B**) Likewise, cumulative survival analyses until all bees had died revealed no significant differences among treatment groups (*mestreg*, *p* > 0.85).

**Figure 2 insects-11-00819-f002:**
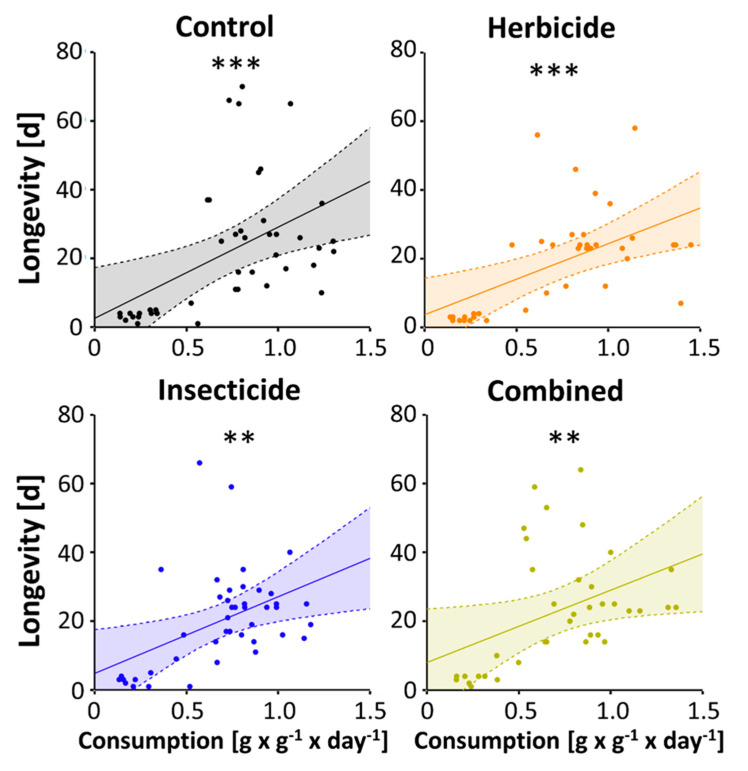
Correlations between consumption and longevity for female adult solitary bees (*Osmia bicornis*). Regardless of the treatment group, a significant positive correlation was observed between longevity and consumption (*lmm*, all *p*’s < 0.003). Each dot represents an individual bee; regression line and 95% confidence interval bands can be seen; significance is represented by the ** (*p* < 0.01) or *** (*p* < 0.001).

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
