# Peer review of "Positive Correlation between Pesticide Consumption and Longevity in Solitary Bees: Are We Overlooking Fitness Trade-Offs?"

_insects, 2020, doi:10.3390/insects11110819_

Round 1
Reviewer 1 Report
Overall, I thought this was a well-written paper with an excellent review of the body of literature. My biggest concern with the paper is that the main conclusion regarding the fitness trade-off between longevity and another fitness measure is not supported by data within the paper. Interestingly, the authors found that increased pesticide exposure leads to increased longevity. The conclusion that was drawn was that this must be caused by a fitness trade-off, but no fitness correlates were measured. Ideally, the paper would be improved by adding data regarding the fitness correlates. If that is not possible, the authors might want to suggest other possible mechanisms for the increased longevity and moderate the hypothesis regarding the fitness trade-off.
Another minor suggestion would be to add more details about how the pesticide consumption was calculated. This seems like an interesting and useful method to many readers.
Minor comments:
Line 17: should be “or combined”
Line 23: remove “the measurement”
Line 29: remove “groups”
Line 35: your statement about the trade-off between longevity and reproduction is not well supported
Excellent introduction. My only suggestion would be to briefly discuss fungicide exposure. You mention it briefly in line 99, but it had not been brought up in comparison to insecticides or herbicide exposure.
Line 101 and 110: It is unclear how you are using the term “sublethal”. On line 101, you are referencing food consumption, but later refer to decreased survival and suppressed consumption.
Line 126: how were individuals exposed to respective treatments? In feeders?
Line 195: It would be helpful to expand on how the amount of exposure was calculated
Table 1: I think this table could be in the supplement
Line 244-246: I do not think you have any data to support this claim
Line 256-257: It would be helpful to explain further the relationship between caloric restriction and immune deficiency. I can see how they might be connected, but it would help the reader to make this connection more clear
Line 301: you mention pesticide exposure not affecting survival. Are there any other studies that found a positive correlation with exposure and survival?
Line 302-304: I agree that, if possible, it would be ideal to add a measure of fitness to this study. Much of your conclusion relies on the hypothesis that there is some fitness trade-off, but you are only showing the effect on longevity not the cost.
Author Response
Overall, I thought this was a well-written paper with an excellent review of the body of literature.
Response: We thank the reviewer for the kind words and highly appreciate the constructive feedback.
My biggest concern with the paper is that the main conclusion regarding the fitness trade-off between longevity and another fitness measure is not supported by data within the paper. Interestingly, the authors found that increased pesticide exposure leads to increased longevity. The conclusion that was drawn was that this must be caused by a fitness trade-off, but no fitness correlates were measured. Ideally, the paper would be improved by adding data regarding the fitness correlates. If that is not possible, the authors might want to suggest other possible mechanisms for the increased longevity and moderate the hypothesis regarding the fitness trade-off.
Response: We fully agree with the reviewer that measuring fitness or at least proxies of fitness would have benefitted this study and potentially shed light on our hypothesis. We also acknowledged this in the discussion as a pitfall of the study (see line 374-379) and future studies are required to address potential trade-offs between survival and fitness. While indeed our hypothesis is speculative, we nevertheless firmly believe we are not overstretching our mandate, as findings from Sandrock et al. 2014 (10.1111/afe.12041) and Stuligross & Williams, 2020 (doi.org/10.1098/rspb.2020.1390) both support the idea of trade-offs (i.e. no significant effect of neonicotinoid exposure on survival yet significant detrimental effects on reproduction). Furthermore, in light of current risk assessment schemes having a strong emphasis on the measurement survival for non-target pollinator species atTier I, such trade-offs could be potentially devastating, as fitness consequences will most likely remain overlooked, especially if no effects on survival are observed. Interestingly, numerous studies exposing solitary bees to field-realistic concentrations of neonicotinoids reveal non-significant results for the endpoint survival (e.g. Abbott et al. 2008; Nicholls et al. 2017; Sgloastra et al. 2018; Azpiazu et al. 2019). In light of these findings, we have all reason to believe that trade-offs must be considered at the earliest Tier I testing phase to appropriately assess the environmental risks of chemicals (as pointed out by Reviewer 2). Given that standard test guidelines are currently under development for solitary bees (e.g. Osmia sp.) we still have sufficient time to incorporate such critical factors for regulatory frameworks and so effectively mitigate the role of chemicals in the ongoing decline of biodiversity.
Another minor suggestion would be to add more details about how the pesticide consumption was calculated. This seems like an interesting and useful method to many readers.
Response: We agree and have now added a line in the methods explaining how this was done (see 175-177).
Minor comments:Line 17: should be “or combined”
Response: Revised as requested (see line 16-18).
Line 23: remove “the measurement”
Response: Agreed. Done (see line 23-24).
Line 29: remove “groups”
Response: Done (see line 29-31).
Line 35: your statement about the trade-off between longevity and reproduction is not well supported
Response: Please see the comment above and we have also softened the wording (see line 33-36).
Excellent introduction. My only suggestion would be to briefly discuss fungicide exposure. You mention it briefly in line 99, but it had not been brought up in comparison to insecticides or herbicide exposure.
Response: We are delighted to read the reviewer considered our introduction as well-written. We are not fully certain what additional information the reviewer would appreciate in the introduction regarding fungicide exposure. While we indeed understand the relevance of fungicide exposure to solitary bees, we believe that additional information specifically on fungicides is redundant as it would be beyond the scope of this experiment. Nevertheless, we have added a recent review published by Belsky and Joshi 2020 (10.3389/fenvs.2020.00081) that provides further details on this topic (see lines 108-111).
Line 101 and 110: It is unclear how you are using the term “sublethal”. On line 101, you are referencing food consumption, but later refer to decreased survival and suppressed consumption.
Response: We agree and have now changed the wording accordingly (see line 119-122).
Line 126: how were individuals exposed to respective treatments? In feeders?
Response: Thank you for pointing this out. Indeed, bees were feed via 5 [ml] syringes and we have now added this information (see line 137-138).
Line 195: It would be helpful to expand on how the amount of exposure was calculated
Response: Agreed. See comment above or line 175-177.
Table 1: I think this table could be in the supplement
Response: We agree and have moved the table to the supplementary information. The caption for the Table can be found at lines 593-598. In addition, we have adjusted all of the references to Table 1 in the main text body to SI Table 1 (see lines 218-222).
Line 244-246: I do not think you have any data to support this claim
Response: Agreed, we provide no data in the present study supporting this claim and thus have emphasized this point in both the abstract and discussion (see lines 33-36; 267-270). In addition, we have removed a misleading sentence from the introduction (see line 108). However, based on previous studies using solitary bees (see Sandrock et al. 2014; Stuligross & Williams 2020) we are convinced that this may indeed be a mechanism explaining our data.
Line 256-257: It would be helpful to explain further the relationship between caloric restriction and immune deficiency. I can see how they might be connected, but it would help the reader to make this connection more clear.
Response: We agree and have now reworded the sentence and added additional information including two references to clarify the connection (see line 281-283).
Line 301: you mention pesticide exposure not affecting survival. Are there any other studies that found a positive correlation with exposure and survival?
Response: To the best of our knowledge we are not aware of any study revealing such data.
Line 302-304: I agree that, if possible, it would be ideal to add a measure of fitness to this study. Much of your conclusion relies on the hypothesis that there is some fitness trade-off, but you are only showing the effect on longevity not the cost.
Response: Indeed, in hinds sight, it would have been ideal to measure fitness or detoxification capacities as mentioned in lines 374-379. Again, we thank the reviewer for the constructive comments which have significantly improved the quality of our manuscript.

Reviewer 2 Report
The authors present a study evaluating individual and synergistic effects of oral exposure to glyphosate and clothianidin in a solitary bee, specifically evaluating sublethal effects. They found a positive correlation between pesticide consumption and longevity, which is explained as a potential trade-off with fitness when exposed to these pesticides, though this is not specifically measured in this study. This is an important consideration in environmental risk assessment of pesticides to ensure that Tier 1 testing uses appropriate experimental endpoints.
While the manuscript is very well written, I find that the dataset itself is really small. The lack of significance is explained as a tradeoff with fitness, though no fitness measurements were made in this study for the bees that were tested. Though this is acknowledged as a pitfall of the study in the discussion on line 301-306, this strikes me as a pretty critical omission that would greatly strengthen the arguments. I cannot recommend this manuscript for publication unless this reproductive data can be included to validate the conclusions.
Specific comments:
line 113: were 300 orchards sampled or 300 O. bicornis? I'm assuming this is individual bees, but it is unclear in the wording.
line 138: 1.5 ng/g of clothianidin is a very low concentration, even by field-realistic standards. The lack of significance in the data could be the result of treatment dose being too low, though this is not addressed in the discussion.
Results section: Typically it is best to write the results section where the significant data is stated first and the non significant data last. This section jumps around between significant and not significant data.
line 195-197: My understanding is that the feeding trials were conducted concurrently. Please clarify if the feeding treatments were assigned before or after determining whether the bees were non-feeders.
Figure 2: It seems to me there are a lot of outlying points. Including R^2 values will help to evaluate the strength of the positive correlations because visually there seems to be a lot of noise in the data.
line 244-246: Were the bees used in this study the same as those used in reference 10? I think it is a stretch to correlate the results of these two data sets if the reproductive tradeoff wasn't validated in the present study.
line 268-269: "the allocation of resources to chemical degradation likely comes at the expense of other functions." I question if you can make such a strong statement based on the results of this study alone, given how low the clothianidin concentration was.
Author Response
The authors present a study evaluating individual and synergistic effects of oral exposure to glyphosate and clothianidin in a solitary bee, specifically evaluating sublethal effects. They found a positive correlation between pesticide consumption and longevity, which is explained as a potential trade-off with fitness when exposed to these pesticides, though this is not specifically measured in this study. This is an important consideration in environmental risk assessment of pesticides to ensure that Tier 1 testing uses appropriate experimental endpoints.While the manuscript is very well written, I find that the dataset itself is really small. The lack of significance is explained as a tradeoff with fitness, though no fitness measurements were made in this study for the bees that were tested. Though this is acknowledged as a pitfall of the study in the discussion on line 301-306, this strikes me as a pretty critical omission that would greatly strengthen the arguments. I cannot recommend this manuscript for publication unless this reproductive data can be included to validate the conclusions.
Response:We thank the reviewer for the constructive comments and suggestions which have inevitably improve our manuscript. Reviewer 2 has raised similar concerns as reviewer 1 and we have explicitly addressed this in a previous response to a comment from reviewer 1. Please see above. Despite the missing data on the potential effects of the tested pesticides on fitness we arenevertheless convinced that the publication of our manuscript would be of significant relevance for a broad range of readers, especially in the field of ecotoxicology and regulatory bodies. We have softened our wording and emphasized that this data is lacking in the manuscript (see: lines 33-36; 267-270)
Specific comments:
line 113: were 300 orchards sampled or 300 O. bicornis? I'm assuming this is individual bees, but it is unclear in the wording.
Response: We agree and have now moved the '(N=300)' immediately after the 'O. bicornis' to resolve the misleading wording (see line 125-127).
line 138: 1.5 ng/g of clothianidin is a very low concentration, even by field-realistic standards. The lack of significance in the data could be the result of treatment dose being too low, though this is not addressed in the discussion.
Response: We have now addressed that the low clothianidin concentration may also explain our findings (see lines 298-305). However, the applied concentration for clothianidin lies well within the range of residues found under natural field conditions (e.g Pilling et al. 2013, doi: 10.1371/ journal.pone.0077193 or Botias et al. 2015, doi:10.1021/acs.est.5b03459; ).
Results section: Typically it is best to write the results section where the significant data is stated first and the non significant data last. This section jumps around between significant and not significant data.
Response: We understand the reviewer's point. However, we opted to report our findings in line with the chronological order of the experiment and measured endpoints.
line 195-197: My understanding is that the feeding trials were conducted concurrently. Please clarify if the feeding treatments were assigned before or after determining whether the bees were non-feeders.
Response: Correct. To determine if a bee was considered a non-feeder or feeder, the bees had to already be exposed to their treatment groups. We have addressed this in lines 134-144 and 165-175 and see no reason for further clarification.
Figure 2: It seems to me there are a lot of outlying points. Including R^2 values will help to evaluate the strength of the positive correlations because visually there seems to be a lot of noise in the data.
Response: We have now added the overall R2 value for all treatment groups (see line 240-243). While the overall R2 value was 0.24 (i.e. indicating much noise), the high-variability data nevertheless revealed a significant positive correlation for all treatment groups (p < 0.003). Subsequently, the model revealed that the predictor variable (i.e. consumption [g]) still provides sufficient information about the response (i.e. age [d]) even though the data points fall relatively far from the regression line. We are aware that correlation is not causation and were thus careful with our wording throughout the entire manuscript (e.g. see line 267-270).
line 244-246: Were the bees used in this study the same as those used in reference 10? I think it is a stretch to correlate the results of these two data sets if the reproductive tradeoff wasn't validated in the present study.
Response: No, the bees in reference 10 (Sandrock et al. 2014) were not the same as used here in this study. We have responded to the concerns regarding the potential trade-off above and kindly ask the reviewer to see the previous comments.
line 268-269: "the allocation of resources to chemical degradation likely comes at the expense of other functions." I question if you can make such a strong statement based on the results of this study alone, given how low the clothianidin concentration was.
Response: Indeed, this statement is speculative and thus we were careful with our choice of wording (i.e. 'may'). Insect detoxification is known to be activated even at low field-realistic concentrations (e.g. Derecka et al. 2013, doi:10.1371/journal.pone.0068191; Christen et al. 2018, doi.10.1021/acs.est.8b01801) and is energetically costly (see Castañeda et al. 2009; du Rand et al. 2015). Subsequently, the allocation of resources to ensure successful detoxification are inevitable. However, as the data revealed no significant difference in consumption, indicating that the required energetic demands were no compensated by increased food intake, the resources for detoxification could have come at the expense of reproduction/fitness (see Harshman et al. 2007, 10.1016/j.tree.2006.10.008; Schwenke et al. 2016 10.1146/annurev-ento-010715-023924). We have also added these references in the discussion (see lines 292-296; 371-372).
We highly appreciate the critical and constructive comments and thank the reviewer for the time and effort.

Round 2
Reviewer 1 Report
The authors have done an excellent job addressing my concerns. My main concern was the lack of data on fitness measurements. Although they were not able to add the data, they have highlighted in both the abstract and discussion the link between survival and fitness by citing previous data. They have also made it clear that this is one step in accessing risk of pesticids.
Reviewer 2 Report
I was not able to see the author comments to Reviewer 1 so am not certain if my concerns were fully addressed.